# Metabolite Profile of Treatment-Naive Metabolic Syndrome Subjects in Relation to Cardiovascular Disease Risk

**DOI:** 10.3390/metabo11040236

**Published:** 2021-04-13

**Authors:** Moritz V. Warmbrunn, Annefleur M. Koopen, Nicolien C. de Clercq, Pieter F. de Groot, Ruud S. Kootte, Kristien E. C. Bouter, Kasper W. ter Horst, Annick V. Hartstra, Mireille J. Serlie, Mariette T. Ackermans, Maarten R. Soeters, Daniel H. van Raalte, Mark Davids, Max Nieuwdorp, Albert K. Groen

**Affiliations:** 1Department of Internal and Vascular Medicine, Amsterdam University Medical Centers, Location AMC, Meibergdreef 9, 1105 AZ Amsterdam, The Netherlands; a.m.koopen@amsterdamumc.nl (A.M.K.); n.c.declercq@amsterdamumc.nl (N.C.d.C.); p.f.degroot@amsterdamumc.nl (P.F.d.G.); r.s.kootte@amsterdamumc.nl (R.S.K.); k.e.bouter@amsterdamumc.nl (K.E.C.B.); a.v.hartstra@amsterdamumc.nl (A.V.H.); m.davids@amsterdamumc.nl (M.D.); m.nieuwdorp@amsterdamumc.nl (M.N.); 2Department of Endocrinology and Metabolism, Amsterdam UMC, Location AMC at University of Amsterdam, Meibergdreef 9, 1105 AZ Amsterdam, The Netherlands; k.w.terhorst@amsterdamumc.nl (K.W.t.H.); m.j.serlie@amsterdamumc.nl (M.J.S.); m.t.ackermans@amsterdamumc.nl (M.T.A.); m.r.soeters@amsterdamumc.nl (M.R.S.); 3Diabetes Center, Department of Internal Medicine, Amsterdam UMC, Location VUMC at VU University Medical Centers, De Boelelaan 1117, 1081 HV Amsterdam, The Netherlands; d.vanraalte@amsterdamumc.nl; 4Amsterdam UMC, Amsterdam Cardiovascular Sciences, VU University, De Boelelaan 1117, 1081 HV Amsterdam, The Netherlands; 5Wallenberg Laboratory, University of Gothenburg, Bruna Stråket 16, SE-413 45 Göteborg, Sweden

**Keywords:** metabolic syndrome, metabolomics, cardiovascular disease, lipolysis, phospholipids

## Abstract

Metabolic syndrome (MetSyn) is an important risk factor for type 2 diabetes and cardiovascular diseases (CVD). This study aimed to find distinct plasma metabolite profiles between insulin-resistant and non-insulin resistant subjects with MetSyn and evaluate if MetSyn metabolite profiles are related to CVD risk and lipid fluxes. In a cross-sectional study, untargeted metabolomics of treatment-naive males with MetSyn (*n* = 132) were analyzed together with clinical parameters. In a subset of MetSyn participants, CVD risk was calculated using the Framingham score (*n* = 111), and lipolysis (*n* = 39) was measured by a two-step hyperinsulinemic euglycemic clamp using [1,1,2,3,3-^2^H5] glycerol to calculate lipolysis suppression rates. Peripheral insulin resistance was related to fatty acid metabolism and glycerolphosphorylcholine. Interestingly, although insulin resistance is considered to be a risk factor for CVD, we observed that there was little correspondence between metabolites associated with insulin resistance and metabolites associated with CVD risk. The latter mainly belonged to the androgenic steroid, fatty acid, phosphatidylethanolamine, and phophatidylcholine pathways. These data provide new insights into metabolic changes in mild MetSyn pathophysiology and MetSyn CVD risk related to lipid metabolism. Prospective studies may focus on the pathophysiological role of the here-identified biomarkers.

## 1. Introduction

Obesity contributes significantly to the global burden of disease and is on the rise. In the past three decades, the incidence of adults with obesity has increased by 27.5% globally [1]. With an increasing body weight beyond a body mass index (BMI) of 25 kg/m^2^, the overall mortality increases drastically (with roughly 30% per five units BMI increase) [2]. Metabolic syndrome (MetSyn) is characterized by cardiovascular risk factors such as increased fasting glucose, dyslipidemia, hypertension, and central obesity, and often develops as a consequence of a high caloric diet and metabolic changes related to obesity [3]. This syndrome increases the chance of developing type 2 diabetes (T2D) and cardiovascular diseases (CVD) [4,5], as well as other diseases including non-alcoholic fatty liver disease (NAFLD) and chronic kidney disease [6,7]. Previous research focusing on the metabolic profile related to insulin sensitivity deterioration already indicated that a noteworthy amount of hepatic and peripheral insulin resistance variation can be explained by metabolite profiles, especially metabolites from the lipid and amino acid super families [8,9]. A systematic review on metabolite profiles in patients with prediabetes and diabetes also found that lipid-associated metabolites such as phospholipids but also amino acid-related metabolites such as aromatic and branched-chain amino acids are associated with the incidence of progressed prediabetes and T2D [10].

Another important aspect of MetSyn progression is insulin resistance in adipose tissue [11,12]. Insulin resistance has been shown to contribute to the lipid triad, which consists of high plasma triglyceride levels, low high-density lipoprotein levels, and the appearance of small dense low-density lipoproteins. This, together with endothelial dysfunction, facilitates atherogenesis [13]. However, it remains to be elucidated what pathophysiological and thus metabolic changes orchestrate the increase in insulin resistance and other metabolic syndrome criteria. The current golden standard to assess insulin resistance is a euglycemic hyperinsulinemic clamp, which provides information about endogenous glucose production suppression and the rate of disposal (Rd), which is the most accurate measure for peripheral insulin resistance at a fixed insulin concentration [14,15].

Hyperlipidemia and in particular high levels of low-density lipoprotein cholesterol (LDL-C) are important drivers of atherogenesis [16]. Yet, although lipids are important, a multitude of other factors play a role as well. The progression of atherosclerosis involves a complex interplay of inflammatory pathways; the reflection of these pathways in the plasma metabolome may be expected. Interestingly, so far, only four plasma metabolites have been firmly indicated to be associated with CVD events in three prospective population-based cohorts; phenylalanine and monounsaturated fatty acid levels were associated with an increased CVD risk. In contrast, omega-6 fatty acids and docosahexaenoic acid levels were associated with a lower CVD risk [17]. Cross-sectional studies indicated a role for trimethylamine N-oxide (TMAO) and recently phenylacetylglycine [18], both metabolites are produced by the gut microbiome and involved in platelet activation [19]. To our knowledge, the direct association between gold standard parameters of glucose handling and CVD risk profiles has not been studied. Therefore, in the present study, we concentrated in a cross-sectional study on the link between insulin resistance and CVD risk.

To obtain insight in early changes in the trajectory of the metabolic syndrome, we selected a homogenous group of men with MetSyn who were treatment-naive. First, the association between metabolites and insulin resistance on both glucose flux as well as lipolysis was determined. Subsequently, the association of plasma metabolites and CVD risk factors was determined. The Framingham score was used to quantify CVD risk. This score predicts the percentage chance of developing a cardiovascular event in the upcoming 10 years.

## 2. Results

### 2.1. Baseline Characteristics

In total, we included 132 men with MetSyn and a BMI of 33.91 kg/m^2^ [31.45, 37.05, median ± IQR]. All included participants were males. In order to find early changes in insulin resistance, we divided participants into insulin-resistant and non-insulin-resistant men based on the rate of glucose disposal (Rd). Subjects with an Rd lower than 37.3 were considered insulin resistant, as previously determined [20]. Men with insulin resistance had a BMI of 34.5 [31.58, 38.69], whereas men without insulin resistance had a BMI of 33.40 [30.82, 35.00]. Age was not different between the groups, but insulin-resistant participants had a higher body weight. There was no difference in blood pressure or fasted plasma glucose levels, but fasting insulin levels were higher in the insulin-resistant group (123.00 pmol/L [93.00, 158.50]) than in the non-insulin-resistant group (69.00 pmol/L [54.75, 87.00]). Inflammatory markers and energy expenditure were also not different between the two groups (Table 1).

### 2.2. Metabolites in Relation to Peripheral Insulin Resistance

To select relevant metabolites that might contribute to the development of insulin resistance or function as a measure for insulin resistance progression, we used a machine learning approach (See Method Section 4.5). The 20 most important metabolites that contributed to distinguishing between peripheral insulin resistance were then compared with Mann–Whitney U tests. Sixteen out of 20 metabolites were significantly different between the groups, and after correction for multiple testing, 14 metabolites showed different abundances between the groups (*q* < 0.05, Appendix A). These metabolites were mainly derived from fatty acids or lysophospholipids (Appendix A). Ten of the most significant metabolites are shown in Figure 1.

### 2.3. Metabolites in Relation to Cardiovascular Disease Risk

As men with metabolic syndrome are at increased risk for CVD, we hypothesized that metabolite profiles in MetSyn show associations to the Framingham risk score. Framingham scores could be calculated in 111 MetSyn participants; in the other MetSyn participants, not all components of the Framingham score were available. Using a q-value < 0.05 as the threshold, 33 metabolites significantly correlated with Framingham score (Appendix A). Five of these metabolites were metabolites that belonged to the phosphatidylcholine pathways of which four were related to LDL-C, none to high-density lipoprotein cholesterol (HDL-C), and two metabolites were related to the phosphatidylethanolamine pathway, of which none were related to LDL-C or HDL-C. The nine metabolites that were most strongly correlated with the Framingham score (*q* < 0.01) are depicted in Figure 2.

To evaluate which metabolites are associated with a high or low Framingham score, we used a cutoff value just below the median Framingham score (12%) to separate the MetSyn participants into two groups, having a high (*n* = 69) or low Framingham score (*n* = 42). The baseline characteristics in these two groups are shown in Appendix A. We used a machine learning approach to classify participants as having a high or low Framingham score based on all annotated (*n* = 917) metabolites. Distinction between the groups using all metabolites was only modestly possible (AUC = 0.66 ± 0.09); however, using only the 20 most predictive metabolites generated reasonable performance of the models (AUC = 0.75 ± 0.09, Figure 2). From these 20 metabolites, 10 were identical to metabolites, which significantly correlated to the Framingham score with univariate Spearman’s rank correlation (Table 2). Framingham score was not directly correlated to the peripheral insulin resistance (*p* = 0.17, rho = −0.13).

The ten metabolites that overlapped between the 20 most predictive features for machine learning classification and univariate correlation to the high and low Framingham score groups were mostly related to lipid metabolism, while two were related to amino acid metabolism.

### 2.4. Metabolites in Relation to Lipolysis

In a subset of 39 MetSyn participants, lipolysis was measured by hyperinsulinemic euglycemic clamp with stable isotopes. The median percent suppression (Ra-supp) of glycerol appearance, a measure for adipose tissue insulin resistance, was 68.1%, and the median Framingham score in this subset of MetSyn participants was 13% (Appendix A), using this as a threshold to compare Ra-supp between men with a higher (*n* = 22) or lower (*n* = 17) Framingham score was not statistically significant with non-parametric comparison (*p* = 0.36). Univariate correlation between all annotated (*n* = 917) metabolites and Ra-supp was not significant after FDR correction. One metabolite was significant after univariate correlation with FDR correction of lipid metabolites (*n* = 451) and Ra-supp, which was the metabolite related to fatty acid metabolism: oleoylcarnitine [C18, *q* = 0.037] (Appendix A). Furthermore, homeostatic model assessment for insulin resistance (HOMA-IR) (*q* < 0.001) and insulin (*q* < 0.01) levels correlated with Ra-supp. A correlation between glucose (*p* = 0.04) and Ra-supp was not significant after multiple testing correction (*q* = 0.15). Other clinical parameters such as LDL-c, HDL-c, BMI, age, and triglycerides did not correlate with lipolysis suppression (Appendix A).

## 3. Discussion

This study showed that men with MetSyn and peripheral insulin resistance have a different metabolic signature compared to MetSyn men without peripheral insulin resistance, and this difference is mainly found in metabolites related to fatty acid metabolism. Interestingly, although insulin resistance is often directly linked to CVD risk, in this group, we did not find a significant correlation between these two parameters. Yet, within the MetSyn group, we could discern a difference between high and low Framingham score. This difference is related to a number of metabolites of varying origin. Although a number of these metabolites were also lipid derived, there was no overlap with the metabolites predicting insulin resistance. Interestingly, only one metabolite was related to insulin-induced inhibition of lipolysis.

In our study, several phospholipids such as phosphatidylcholine (PC) and phosphatidylethanolamine (PE) were found to be associated with the Framingham score. Phospholipids have already been described to be associated with MetSyn components; for example, PC-related metabolites are associated with hypertension and both PC and PE-related metabolites are associated with dyslipidemia [21]. It has been suggested that the PC:PE ratio has significant implications for metabolic homeostasis and that obesity with underlying disturbances of fatty acid metabolism disrupts this balance [22]. In fact, a decreased PC:PE ratio is associated with obesity [23], NAFLD [24], prediabetes, as well as type 2 diabetes in human studies [25]. A spectrum of diseases with common characteristics such as low-grade inflammation is commonly referred to as cardiometabolic diseases (CMD) [26]. PC and PE are important components of the plasma membrane. However, PC is found in higher levels on the outer cell membrane, whereas PE is usually mainly found in the inner membrane. Disruption of this distribution has been suggested to change the permeability of membranes for cytokines as well as affect membrane potential [22]. However, as of now, the relation between phospholipid levels in the plasma membrane and blood plasma is still unclear. Further study is required to unravel the molecular mechanism underlying the relation between the PC:PE ratio and metabolic diseases. Interestingly, both metabolites from PC and PE metabolism were positively associated with Framingham score (Appendix A). Previous research has already indicated that PC and PE related metabolites are distinctive for individuals with or without MetSyn [27] but not in many studies.

Most metabolites found to be different between MetSyn indivduals with and without insulin resistance showed a higher abundance in the insulin resistance group, and most of these metabolites were related to fatty acid metabolism. Participants in our study had an increased BMI, and it is already known that free fatty acid levels are higher in obesity [28], which can cause mitochondrial dysfunction [29]. By using the golden standard for insulin resistance, our findings suggest that fatty acid metabolism is already disturbed in men with treatment-naive MetSyn independent of fasting glucose.

In contrast to most changed metabolites in peripheral insulin resistance, glycerophosphorylcholine had a higher abundance in the non-insulin resistant group. This metabolite has been shown to have beneficial or detrimental effects on cell survival in primary rat cardiomyocytes, depending on the time of exposure [30]. In addition, this metabolite has been shown to reduce oxidative stress in rat liver cells by preserving mitochondrial complex 1 function [31] and can also protect against cardiac ischemia–reperfusion damage in vivo [30]. As particularly peripheral insulin resistance in MetSyn contributes to the development of insulin resistance [32], this metabolite might also protect against oxidative stress in MetSyn, which then ameliorates the progression of insulin resistance.

We found that several (polyunsaturated) fatty acids correlated with Framingham CVD risk, with the exception of eicosenedioate, which correlated inversely with CVD risk (Table 2, Appendix A). Two metabolites related to polyunsaturated fatty acids, heneicosapentaenoate and docosahexaenoylcarnitine, were positively correlated with CVD risk as well as being predictive in machine learning models. This might seem counter intuitive, as polyunsatured fatty acids have been shown to have anti-inflammatory effects and improve insulin sensitivity in mice [33]. Since heneicosapentaenoic acid can also be incorporated into triacylglycerol as well as phospholipids, it was considered unlikely to exert beneficial biological effects [34].

In our study, only one metabolite was correlated with the suppression of lipolysis. This metabolite, oleoylcarnitine, belongs to the group of long-chain acylcarnitines, which is a group that previously already has been associated with insulin resistance [8,9]. Obese and type 2 diabetic men were shown to have elevated levels of acylcarnitines [35]. It has been reported that a high influx of fatty acids to muscles as well as isolated mitochondria promotes incomplete β-oxidation of long chain fatty acids, which could result in an accumulation of acylcarnitines, inducing oxidative stress, mitochondrial dysfunction, and insulin resistance [36,37,38], underlining the importance of this metabolite in obesity and insulin resistance.

We initially hypothesized that lipid flux and thus also lipolytic changes in MetSyn participants would relate to clinical parameters such as LDL-c and triglycerides. However, only HOMA-IR and glucose correlated to lipolysis suppression in our study. This could be due to the fact that more profound changes in lipid metabolism have to manifest before a distinct metabolite profile can be identified. MetSyn participants in our study had a high BMI but relatively low LDL-c and HbA1c; therefore, lipid metabolism in this group might still function relatively well.

A major limitation of this study is the cross-sectional design. To understand what drives the deterioration of metabolic parameters such as insulin resistance and dyslipidemia, it is also necessary to follow subjects prospectively. In addition to this, it is important to validate the differences in metabolic profile found between the sub groups of MetSyn subjects and look at potential gender-related effects. It would be interesting to compare the non-insulin-resistant men with healthy men; this could be assessed in the future by adding an additional control group of healthy age matched study participants. In addition to this, to better understand the physiology of insulin resistance and the relation with cardiovascular disease risk, more measurement of metabolic fluxes are necessary as well as hard outcomes for cardiovascular diseases.

This study can be seen as an initial step to better understand the onset of the metabolic syndrome and the features that induce CVD. Future prospective studies should evaluate if the metabolites found in this study are indeed important for MetSyn progression and cardiovascular risk. When confirmed, the molecular mechanism should be assessed in in vitro experiments, and subsequent implementation in clinical studies, with for example, dietary or pharmacologic interventions studies, would pave the way for new therapeutic strategies to target MetSyn progression toward T2DM and CVD.

## 4. Materials and Methods

### 4.1. Study Design and Population

In a cross-sectional study design, Caucasian males meeting three or more of the National Cholesterol Education Program (NCEP) Adult Treatment Panel (ATP III) criteria for metabolic syndrome such as waist circumference ≥ 102 cm, increased blood pressure (>130/85 mmHg), fasting blood glucose ≥ 5.6 mmol/L, triglycerides ≥ 1.7 mmol/L, and high-density lipoprotein (HDL) cholesterol < 1.0 mmol/L were included for the MetSyn group [39]. Other inclusion criteria for the MetSyn group was BMI > 28 kg/m^2^, no medication use, and no other comorbidities. Exclusion criteria for this study were a history of cholecystectomy, use of any type of medication, and cardiovascular diseases. A comprehensive overview of inclusion and exclusion criteria has been previously published [40]. The institutional review board of the Amsterdam University Medical Center, location AMC approved all study procedures, which were in compliance with the declaration of Helsinki. The study was registered at the Dutch trial register (NTR4913 and NTR2705). Volunteers provided written informed consent.

### 4.2. Clinical Parameters and Framingham Score

The Framingham risk score was calculated in 111 MetSyn participants based on age, sex, total cholesterol, HDL-c, systolic blood pressure, blood pressure treatment, smoking status, and diabetes status, as previously described [41]. Biochemistry and clinical parameters such as fasting glucose, HbA1c, LDL-c, and HDL-c were analyzed with routine laboratory measurements as previously described [9]. Peripheral insulin resistance was defined as having a Rd lower than 37.3, as previously described [20].

### 4.3. Metabolite Analysis

Ethylenediaminetetraacetic acid (EDTA) plasma samples were collected from overnight fasted study subjects for both groups. Untargeted metabolomics analysis using ultra high-performance liquid chromatography with tandem mass spectrometry (UPLC-MS/MS) was performed by Metabolon (Durham, NC, USA), as previously described [42]. In total, 917 annotated and 236 unannotated metabolites were measured. Raw abundance data were normalized to control for differences between measurements, and the median of every metabolite was rescaled for all samples to 1. If data were missing, for example due to measurements falling below the detection limit, 50% of the minimum observed value from the respective metabolite through all samples was used to impute missing values.

### 4.4. Two-Step Hyperinsulinemic Euglycemic Clamp and Lipolysis

Insulin sensitivity was measured by a two-step hyperinsulinemic clamp in all subjects (*n* = 132). In a subset of the Metsyn group (*n* = 39), lipolysis inhibition was measured by a two-step hyperinsulinemic clamp using the stable isotope [1,1,2,3,3-^2^H_5_] glycerol (99% enriched; Cambridge isotopes, Andover, MA, USA), in 20% glucose and 1% [6,6-^2^H2] glucose and insulin (Actrapid; Noco Nordisk Farma, Alphen aan de Rijn, The Netherlands), which was administered through one catheter and another was used to obtain blood samples. Two hours before starting (t = −2) with the clamp, the glycerol isotope was infused (500mg). After the initial two hours, insulin infusion was started at a rate of 20 mU m^−2^ min^−1^, and every 10 min, plasma glucose was measured by a glucose analyzer (YSI 2300 Stat Plus Glucose Lactate Analyzer, YSI Life Sciences, Yellow Springs, OH, USA) and 20% glucose enriched at a variable rate was infused to consistently maintain glucose at 5 mmol/L. At start of the clamp and after 2 and four hours, blood samples were obtained to measure glycerol enrichment. Glucose and lipid flux or rate of glycerol appearance suppression was calculated with the modified Steele equations for (non-) steady-state measurements [43,44]. Study subjects were not hospitalized; clamps were performed in an outpatient research facility.

### 4.5. Statistical Analysis and Machine Learning Models

Mann–Whitney U testing was performed to compare metabolites between different groups, and univariate correlations (Spearman’s rank) were calculated to find associations between outcomes and metabolites. For both methods, false discovery rate (FDR) correction was applied to control for multiple comparisons.

For the machine learning approach to classify groups based on metabolites, XGBoost (version 0.90) was used with implementation of gradient boosted trees. To ensure robustness of results, a nested cross-validation system was used to prevent overfitting. Predictive models followed 100 structured iterations. Data were split into a training set (80% of participants) and into a test set (20% of participants) in every iteration. Five-fold cross-validation was performed in the training data to optimize and fit model hyperparameters. Subsequently, the optimized models were tested on the training data, and the area under the curve was recorded for every iteration and the mean AUC was used as final model result. As a sanity check, two random variables were added to the predictor data in each iteration to prevent irrelevant features to be identified as important.

Statistical analyses were performed in R (version 3.6.1) and ggplot2, fmsb, and mixOmics packages were used for visualization and statistical analyses. Machine learning models were performed with Python (version 3.7.4) and the following packages: pandas (version 0.25.1), scikit-learn (version 0.21.2) and numpy (version 1.16.4).

## 5. Conclusions

A cohort of 14 metabolites predicts insulin resistance in MetSyn men. Interestingly, these specific metabolites do not predict CVD risk in these subjects. The most distinct metabolites related to peripheral insulin resistance were metabolites from fatty acid metabolism and glycerophosphorylcholine. The most important metabolites related to CVD risk were related to androgenic steroids, fatty acid, phosphatidylethanolamine, and phophatidylcholine metabolism. Overall, the most distinctive metabolites were related to the lipid super pathway, and prospective studies are needed to evaluate the role of found metabolites in the onset and progression of the metabolic syndrome.

## Figures and Tables

**Figure 1 metabolites-11-00236-f001:**
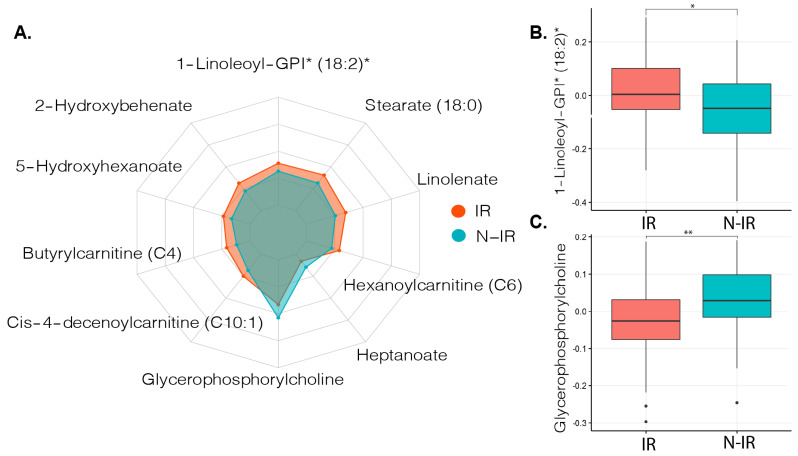
Metabolite profiles related to insulin resistance in metabolic syndrome (MetSyn). (**A**). Radarchart depicting the ten most distinct metabolites between MetSyn participants with and without peripheral insulin resistance. * Metabolite ID estimated based on molecular weight and presented as relative abundance. (**B**). Boxplot depicting median and interquartile ranges between metabolite abundance for 1-Linoleoyl-GPI (18:2), Mann–Whitney U test with false discovery rate (FDR) correction *q* = 0.012. * *q* < 0.05. (**C**). Boxplot showing median and interquartile ranges between metabolite abundance for Glycerophosphorylcholine Mann–Whitney U test with FDR correction *q* = 0.004. ** *q* < 0.01, IR: insulin resistant (*n* = 92) based on Rd < 37.3 μmol kg^−1^ min^−1^, N-IR: non-insulin resistant (*n* = 40) based on peripheral insulin resistance Rd ≥ 37.3 μmol kg^−1^ min^−1^.

**Figure 2 metabolites-11-00236-f002:**
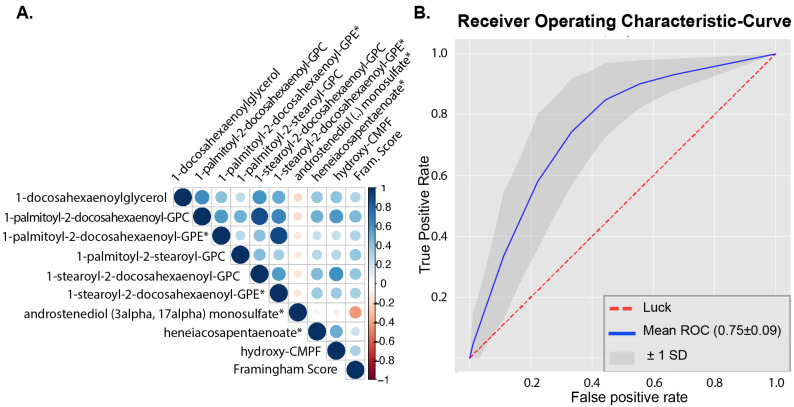
Metabolites related to Framingham score. (**A**). Correlogram between all annotated metabolites (*n* = 917) and Framingham score. Metabolites shown were significantly correlated to Framingham score using Spearman’s rank correlation with FDR correction *q*-value < 0.01. * metabolite was also in top 20 predictors from machine learning classification between healthy and MetSyn; for a complete overview of overlapping metabolites, see Table 2 For subpathway and superpathway annotations, see Appendix A. (**B**). Receiver operating characteristic (ROC) curve of top 20 metabolites selected from initial machine learning model with all annotated metabolites (*n* = 917).

**Table 1 metabolites-11-00236-t001:** Baseline characteristics of participants. Data are depicted as mean (SD) or median [InterQuartile Range]. CRP: C-reactive protein, BP: blood pressure, HDL-C: high-density lipoprotein cholesterol, LDL-C: low-density lipoprotein cholesterol; REE: resting energy expenditure; IR: insulin resistant; N-IR: Non-insulin resistant.

	Overall	IR (Rd < 37.3)	N-IR (Rd ≥ 37.3)	*p*-Value
*n*	132	92	40	
Age (years)	53.83 (9.38)	53.61 (10.31)	54.35 (6.89)	0.678
BMI (kg/m^2^)	33.91 [31.45, 37.05]	34.50 [31.58, 38.69]	33.40 [30.82, 35.00]	0.024
Weight (kg)	115.40 [102.10, 124.62]	117.70 [105.45, 130.32]	108.45 [99.95, 118.17]	0.003
Syst (mmHg)	143.61 (18.20)	144.59 (18.63)	140.72 (16.85)	0.326
Diast (mmHg)	89.45 (11.11)	90.58 (10.93)	86.14 (11.15)	0.063
Gluc (mmol/L)	5.72 (0.66)	5.74 (0.70)	5.68 (0.55)	0.599
Insulin (pmol/L)	109.00 [70.75, 141.75]	123.00 [93.00, 158.50]	69.00 [54.75, 87.00]	<0.001
Rd (μmol kg^−1^ min^−1^)	31.37 [22.65, 40.02]	27.00 [19.86, 33.01]	48.10 [41.08, 55.45]	<0.001
HbA1c (mmol/mol)	39.00 [36.00, 41.00]	39.00 [36.00, 42.00]	38.50 [37.00, 40.75]	0.58
HOMA-IR	3.70 [2.50, 5.16]	4.29 [3.19, 5.46]	2.55 [1.90, 3.05]	<0.001
Total chol (mmol/L)	5.00 [4.59, 5.89]	5.09 [4.56, 5.85]	4.90 [4.63, 6.02]	0.831
LDL (mmol/L)	3.30 [2.70, 4.10]	3.26 [2.70, 4.00]	3.38 [2.66, 4.15]	0.974
HDL (mmol/L)	1.08 [0.93, 1.23]	1.04 [0.93, 1.21]	1.10 [0.96, 1.33]	0.268
Trig (mmol/L)	1.40 [1.12, 1.79]	1.42 [1.16, 1.80]	1.23 [1.10, 1.66]	0.097
ALAT (U/L)	33.00 [26.00, 41.00]	34.00 [27.00, 43.00]	31.00 [22.50, 36.25]	0.022
CRP (mg/L)	2.00 [1.30, 4.35]	2.20 [1.37, 4.70]	2.00 [1.05, 3.80]	0.544
Leuko (10^E^9/L)	6.01 (1.41)	6.02 (1.37)	5.95 (1.57)	0.845
REE (kcal/day)	1939.00 [1804.00, 2190.50]	1952.00 [1806.00, 2246.70]	1924.00 [1760.00, 2083.25]	0.172

**Table 2 metabolites-11-00236-t002:** Metabolites related to high vs. low Framingham score. Metabolites overlapping between top 20 machine learning features to classify between high and low Framingham and univariate correlation to Framingham score. Low Framingham is defined as having a Framingham score below 12%, and 12% or higher is defined as a high Framingham score. This distinction is based on an optimal cutoff around the median Framingham score, which was 12%. Median and interquartile ranges for low and high Framingham score from metabolite abundances. * Metabolite ID estimated based on molecular weight and presented as relative abundance.

Biochemical Annotation	Subpathway Annotation	Super-Pathway Annotation	Low Framingham (*n* = 42, Median [IQR]))	High Framingham (*n* = 69, median [IQR])
Eicosapentaenoate (EPA)	Long-Chain Polyunsaturated Fatty Acid (n3 and n6)	Lipid	0.85 [0.69, 1.20]	1.19 [0.91, 1.65]
N-acetyltyrosine	Tyrosine Metabolism	Amino Acid	0.93 [0.82, 1.17]	1.22 [0.88, 1.57]
1-stearoyl-2-docosahexaenoyl-GPE (18:0/22:6) *	Phosphatidylethanolamine (PE)	Lipid	0.79 [0.60, 1.05]	1.21 [0.77, 1.50]
Docosahexaenoylcholine	Fatty Acid Metabolism (Acyl Choline)	Lipid	0.81 [0.62, 1.28]	1.14 [0.89, 1.58]
Heneicosapentaenoate (21:5n3)	Long-Chain Polyunsaturated Fatty Acid (n3 and n6)	Lipid	0.26 [0.26, 0.60]	0.71 [0.26, 1.70]
Androstenediol (3alpha, 17alpha) monosulfate (2)	Androgenic Steroids	Lipid	1.16 [0.91, 1.69]	0.81 [0.65, 1.02]
3-carboxy-4-methyl-5-propyl-2-furanpropanoate (CMPF)	Fatty Acid, Dicarboxylate	Lipid	0.68 [0.26, 1.63]	1.61 [0.62, 2.62]
1-palmitoyl-2-docosahexaenoyl-GPE (16:0/22:6) *	Phosphatidylethanolamine (PE)	Lipid	0.74 [0.58, 1.19]	1.23 [0.78, 1.45]
1-palmitoyl-2-stearoyl-GPC (16:0/18:0)	Phosphatidylcholine (PC)	Lipid	0.96 [0.84, 1.02]	1.04 [0.93, 1.12]
lanthionine	Methionine, Cysteine, SAM, and Taurine Metabolism	Amino Acid	0.23 [0.23, 0.38]	0.69 [0.23, 1.22]

## Data Availability

The data presented in this study are available in the article.

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
