# Peer review of "Metabolite Profile of Treatment-Naive Metabolic Syndrome Subjects in Relation to Cardiovascular Disease Risk"

_metabolites, 2021, doi:10.3390/metabo11040236_

Round 1

Reviewer 1 Report

The authors aimed to investigate distinct plasma metabolite profiles between insulin resistant and non-insulin resistant MetSyn participants and evaluate if MetSyn metabolite profiles are related to CVD risk and lipid fluxes. Overall, the study design is well described and structured. However, there are several weaknesses in this study which need to be addressed.

-It is of special interest to investigate specific lipid metabolomic biomarkers that could be used for identification of metabolic syndrome subjects with peripheral insulin resistance and with higher cardiovascular risk. Receiver-operating characteristics (ROC) analysis should be calculated to examine the discriminatory ability of the identified metabolites as biomarkers to discriminate between metabolic syndrome subjects with peripheral insulin resistance in clinical practice by the area under the curve (AUC). AUC has been reported to be a global indicator of diagnostic performance.

-A control group of healthy subjects in required in this study in order to compare the non-insulin resistant individuals with healthy subjects.

- The language of the present study must be revised. There are several inconvenient sentences, especially in the objectives, results and discussion part. It should be more clear.

-The authors should make clear in the introduction what the originality of your study is, and what it adds to new results. I would advise the authors to add very few lines in introduction section in order to highlight even more the need to identify a metabolic profile of metabolic syndrome subjects in relation to CVD risk.

-Did authors use a sample size calculator? Sample size should be estimated ahead of time to insure that there is enough power to produce the outcome.

- In general, in the discussion it is necessary that authors can do a succinct literature review on the results of previous studies and compare with the results obtained in this study.

- This study showed that individuals with MetSyn and peripheral insulin resistance have a different metabolic signature than MetSyn individuals without peripheral insulin resistance and this difference is mainly found in metabolites related to fatty acid metabolism. Please discuss the potential underlying mechanisms and the relevance of these findings.

- I think one point that needs to be made more clear is that this is a cross-sectional study and it is unknown whether many of these subjects will develop peripheral insulin resistance in the future. Therefore, a future longitudinal analysis is necessary to determine temporal relationship between changes in metabolic profile and the development of peripheral insulin resistance.

-Since the study cohort only comprised males, this could imply that these findings may not be generalizable to women.

- Please highlight the clinical relevance of this study.

- The conclusion of this study must be more precise, please consider re-writing it.

Reviewer 2 Report

  1. Your individuals were mainly obese (BMI> 30), according to the medians ad IQR in tables 1 and S3. Why did you use the inclusion criteria BMI =28 ? Wouldn't be better to include obese men (BMI> 30) and add this information to the topic of this manuscript.
  2. The word “individuals" should be change to "men" or "obese men" in this manuscript.
  3. Naïve – typo
  4. Line 284

„In a subset of the Metsyn group (n = 39) lipolysis inhibition was measured”, but the  rate of glucose disposal was measured in 132 individuals (Table 1), Please include this information in method section. Please include information that individuals of these study were hospitalized to perform the two-step hyperinsulinemic euglycemic clamp and lipolysis, (If yes?). 

  1. You found the difference in e.g. BMI between IR and N-IR, I think that your metabolites results should be adjusted for BMI. 
  2. You used cut-off for Rd (37,3). Please precise the reference for this cut-off or the way of calculation.
  3. It would be interesting to calculate the ROC curve for nine metabolites, which had the best correlation with Framingham score.
  4. Table 2 please include p values for the differences in metabolites
  5. Why did you use median 12% (your median is 13% in table S3?). You could use additionally the cut-off value from the established reference e.g. N. John Bosomworth, Canadian Family Physician April 2011, 57 (4) 417-423 “Risk is considered low if the FRS is less than 10%, moderate if it is 10% to 19%, and high if it is 20% or higher”
  6. Line 262 “ Individuals in the control group had to be lean (BMI < 25 kg/m2 262 ,) healthy Caucasian males” Did you use control group in your statistical analysis? Two-step hyperinsulinemic euglycemic clamp, lipolysis and Framingham score were done in MetS group?
  7.  line 176: “we did not find a significant correlation between these two parameters.” Did you calculate the correlation between Rd and CVD risk score?
  8. Line 298:  “Mann-Whitney U testing was performed to compare metabolites between different groups”. But you show the mean and SD (t- student test) in tables?

Reviewer 3 Report

Summary:  Paper reports on a study that compared the plasma metabolic profile of insulin resistant and non-insulin resistant patients with metabolic syndrome (MetSyn) in an effort to identify metabolites that contribute to cardiovascular risk in patients with type 2 diabetes.  Results showed little correlation between metabolites associated with insulin resistance and those associated CVD risk.  There were, however, difference in metabolites between MetSyn patients with and without insulin resistance.  In the combined group the major metabolites associated with CVD risk were those in the androgenic steroid, fatty acid and phosphatidylcholine pathways.   The paper concludes that these data provide insights into the metabolic changes that accompany MetSyn as well as those that contribute to CVD risk.

Specific comments:

  • The patients studied were males – would similar results be expected for females? Please include this as a study limitation in the Discussion and discuss briefly.
  • Please include waist circumference in table 1 since that is one of the criteria for defining metabolic syndrome. Also include units for CRP, Leuko and REE.
  • Page 10. In the description of the clam studies please indicate the dose of the glycerol isotope that was administered for the lipolysis measurement studies.  If there is a reference for this method please indicate.
  • Table 2. Please indicate that the units for the values in these table is relative abundance.
  • There are some errors in the text that should be corrected. Line 103, repeated “and”. Line 142, “shwon” should be “shown”.  Table 1”Hba1c” should be “HbA1c” and “Totchol” should be “Total Chol”.  Table 2, close the parentheses in the end of the second column of the first row.

Reviewer 4 Report

The paper by Warmbrunn et al. profiles metabolites of MetSyn individuals. The association between metabolites and insulin resistance on both glucose flux as well as lipolysis was determined. Then the association of plasma metabolites and CVD risk factors was determined. Framingham score was applied to quantify CVD risk and statistical analysis and machine-learning models were performed.

As it is already known from former studies lipids are a promising molecular tool for clinical and therapeutically management of metabolic syndrome. This study gathers information on specific lipid classes and subclasses. The authors found that overall the most distinct metabolites were related to the lipid super pathway. However, prospective studies and more in detail studies including also healthy individuals are needed as authors also mention in there manuscript.

Generally, the paper is well written and organized. However, I recommend to improve table 1 as it is not clearly arranged and the axis labels of figure 1 (B and C) and figure 2 (A and B).

Round 2

Reviewer 1 Report

I suggest accept in present form

Reviewer 2 Report

Thank you very much for responding to  my comments.

Minor comment.

1. The results of Fig.S1 and S2 could be shortly described in the results section.

This manuscript is a resubmission of an earlier submission. The following is a list of the peer review reports and author responses from that submission.

Round 1

Reviewer 1 Report

Metabolic syndrome is a complex group of conditions that contribute to a high risk of blood vessels related pathologies, such as stroke and coronary heart diseases. Metabolic studies in this field have increased tremendously in an effort to characterize the pathology and allow the development of targeted treatments to what is still a largely uncharacterized syndrome. The authors with this study aimed at identifying metabolites that would allow to profile patients with metabolic syndrome. Although the aim is very important, the design of the study and the contribution to the field is questionable. The study compares a group of healthy individuals of young age to a group of obese individuals with metabolic syndrome that are older. Some parameters analyzed are strongly influenced by age and results of the identified metabolites may have no significant differences when comparing individuals of same age, therefore losing their potential as clinical markers. Other important factor, the sex of the individuals, was left out. The study as it is fails to provide a valuable profiling for individuals with metabolic syndrome. 

Author Response

Reply to Reviewer 1

Metabolic syndrome is a complex group of conditions that contribute to a high risk of blood vessels related pathologies, such as stroke and coronary heart diseases. Metabolic studies in this field have increased tremendously in an effort to characterize the pathology and allow the development of targeted treatments to what is still a largely uncharacterized syndrome. The authors with this study aimed at identifying metabolites that would allow to profile patients with metabolic syndrome. Although the aim is very important, the design of the study and the contribution to the field is questionable. The study compares a group of healthy individuals of young age to a group of obese individuals with metabolic syndrome that are older. Some parameters analyzed are strongly influenced by age and results of the identified metabolites may have no significant differences when comparing individuals of same age, therefore losing their potential as clinical markers [1]. Other important factor, the sex of the individuals, was left out [2]. The study as it is fails to provide a valuable profiling for individuals with metabolic syndrome. 

  1. We thank the reviewer for the critical remarks and have performed additional analyses to address the issues raised. To address the possible correlation between age and identified metabolites we performed a spearman correlation test between age and the discussed metabolites in table 2 (which are distinctive between MetSyn and healthy, metabolites from Table 3 are the results from a separate analysis performed only in MetSyn participants). This analysis showed that when MetSyn and healthy are analyzed separately, only one metabolite in the MetSyn group (from androgen metabolism; androstenediol [3alpha, 17alpha] monosulfate [2]) is associated with age, all other metabolites are not correlated to age in both groups. This suggests that the discerning property of these metabolites, which were identified by machine learning, cannot be explained by only the age differences between the groups. See also line  147-154 and two new tabs in Table S1. Note that the age range in both groups is substantial making the correlation test valid. To make the age difference in the groups more insightful, age ranges have been added to Table 1.
  2. In this study only males have been included (see also section Study design and population in the Materials and Methods section line 329), To stress this point we added information on the sex of included study subjects in line 34 and 90-91

Reviewer 2 Report

This is a well written paper. My major concern with this paper is the lack of age matching with the control group being significantly younger than the test group. Hence findings such as the increased androgen metabolites being higher in the controls would be expected. This is evident in the heat map where the androgens were more highly associated in the controls but much less in the test group. These issues need to be discussed in more detail and listed in the study limitations.

However an additional assessment of the association between the androgens and the major metabolites against age as was done with the Framingham score could be contemplated to assess the androgen relationships with the other metabolites. This would give the authors additional evidence and improve their ability to understand the influence of the androgens upon the other metabolites and their potential role in metabolic syndrome development and allow further more detailed discussion of the findings. This analysis could be done using the whole group data and then comparing the correlations within the controls with that of the test subjects.

Author Response

Reply to Reviewer 2

This is a well written paper. My major concern with this paper is the lack of age matching with the control group being significantly younger than the test group. Hence findings such as the increased androgen metabolites being higher in the controls would be expected. This is evident in the heat map where the androgens were more highly associated in the controls but much less in the test group. These issues need to be discussed in more detail and listed in the study limitations.[1]

However an additional assessment of the association between the androgens and the major metabolites against age as was done with the Framingham score could be contemplated to assess the androgen relationships with the other metabolites.[2] This would give the authors additional evidence and improve their ability to understand the influence of the androgens upon the other metabolites and their potential role in metabolic syndrome development and allow further more detailed discussion of the findings. This analysis could be done using the whole group data and then comparing the correlations within the controls with that of the test subjects.[3]

  1. We agree with the reviewer that elaboration of findings on possible confounding relations with age are necessary. As also discussed in our reply to reviewer 1 we therefore performed additional analyses and elaborated on this subject in the discussion. Our additional correlation tests between age and the nine metabolites listed in table 2 show that when groups are analyzed separately, only one metabolite in the MetSyn group (from androgen metabolism) is associated with age, suggesting that found differences in metabolites between groups are not explained by age only. See also results from this analysis in newly added tabs in Table S1 and line 147-154. In our limitations sections we already discussed the differences in age (line 310-314), but we agree that more elaboration would be beneficial for our manuscript. We therefore elaborated in the discussion on this, see line 314-316.
  2. Our new analysis shows that all but one metabolite in the MetSyn group are not correlated to age (Table S1), it is therefore unlikely that all other metabolites are correlated to androgenic metabolites (which overall do not correlate with age) and that differences found between the groups are explained by the age difference, we therefore did not further evaluate every metabolite separately to the three androgenic metabolites. However, to allow correct insights into age ranges of studied groups, we added the age ranges in Table 1.
  3. We thank the reviewer for this constructive idea, as discussed in 1., we have performed this additional analyses and found that most (androgenic) metabolites are not related to age. This is discussed in the results section in line 147-154 and in the discussion line 288-291 and 314-316.

Round 2

Reviewer 1 Report

Corrections performed by the authors improved the quality of the manuscript, but the problem of age range persist. The MetSyn group even though has also youger patients that match control individuals, has also much older patients, has revealed by the median age. The fact that age influences only one metabolite in the MetSyn group, does not reveal how these metabolites would change if the control individuals would have macted age. There could be metabolites that for older patients lose their predictive value as they could change as well in control individuals. The addition of more control individuals with older ages is instrumental.

Author Response

We are pleased that the reviewer considers our revised manuscript improved. Reviewer 1 however is still not happy with the difference in median age between our Metsyn and Control group. We agree that our choice of two groups with a difference in median age is unusual but as stated in the manuscript we chose for this approach to maximize the power to find differences between healthy and metabolic syndrome subjects. Interestingly, the differences between these very different groups of subjects were relatively small and a refined machine learning procedure was necessary to obtain adequate prediction potential. Since, indeed age was a potential confounding factor we analyzed in our revised manuscript whether age associated with the concentration of the metabolites governing the predicted difference between the MetSyn and Control group. Of the nine metabolites in two groups, thus 18 correlation tests, only one was associated with age. We do not agree with reviewer 1 that the correlation of this one metabolite jeopardizes the appropriateness of our study design and we feel that adding novel subjects to our Control group is not going to alter our conclusion.

Reviewer 2 Report

On figure 2b there is a misspelling of N-acetyltyrosine (you have N-Acetyltyaine).

Author Response

We thank the reviewer for the positive evaluation of our manuscript and have corrected the mistake in Fig. 2b